Resource

# Genetic structure and diversity of the *rfb* locus of pathogenic species of the genus *Leptospira*

Leonardo CA Ferreira[1], Luiz de FA Ferreira Filho[2], Maria Raquel V Cosate[3], Tetsu Sakamoto[1]

**Leptospirosis is caused by pathogenic strains of the genus *Leptospira* and is considered the most widespread zoonotic bacterial disease. The genus is characterized by the large number of serology variants, which challenges developing effective serotyping methods and vaccines with a broad spectrum. Because knowledge on the genetic basis of the serological diversity among leptospires is still limited, we aimed to explore the genetic structure and patterns of the *rfb* locus, which is involved in the biosynthesis of lipopolysaccharides, the major surface antigen that defines the serovar in leptospires. Here, we used genomic data of 722 pathogenic samples and compared the gene composition of their *rfb* locus by hierarchical clustering. Clustering analysis showed that the *rfb* locus gene composition is species-independent and strongly associated with the serological classification. The samples were grouped into four well-defined classes, which cluster together samples either belonging to the same serogroup or from different serogroups but sharing serological affinity. Our findings can assist in the development of new strategies based on molecular methods, which can lead to better tools for serological identification in this zoonosis.**

## Introduction

Leptospirosis is a zoonotic bacterial infectious disease that affects both humans and commercially relevant animals. The disease is widely distributed geographically and transmitted through direct or indirect exposure to the urine of infected animals, representing a public health challenge in regions with heavy rainfall, floods, and poor socioeconomic conditions (Faine et al, 1999). The causative agent of the disease is the pathogenic species of the genus *Leptospira* (Phylum Spirochaetes), a highly motile gram-negative bacterium owing to its periplasmic flagella (Levett, 2001; Ko et al, 2009). It is estimated that the disease is responsible for over one million cases and about 60,000 deaths annually worldwide (Costa et al, 2015). However, the actual numbers are potentially higher because of its multiform and nonspecific presentation, which leads to misdiagnosis with other diseases such as malaria, hepatitis, and, especially, viral hemorrhagic fevers such as dengue (Lindo et al, 2013; Wysocki et al, 2014).

Until 1989, leptospires were divided into two species, *Leptospira interrogans* and *Leptospira biflexa*, with pathogenic strains represented exclusively by the species *L. interrogans* and saprophytes by *L. biflexa* (Levett, 2001). Later, with the advances in diagnostic methods and sequence technologies, *Leptospira* samples have been classified using two distinct approaches: the serological classification, which is based on their antigenic characteristics called serovar (srv), and taxonomic classification, which is based on the DNA sequence similarity. It is interesting to note that the two forms of classification do not correspond to each other and each one is used for different purposes. Serological classification is important for epidemiological studies of disease and vaccine development (Adler & de la Peña Moctezuma, 2010), whereas taxonomic classification allows us to investigate the evolutionary relationship between samples of the genus (Vincent et al, 2019).

The number of *Leptospira* species described so far sums up to 72 species (Korba et al, 2021; Fernandes et al, 2022; Dos Santos Ribeiro et al, 2023). Studies on the evolution and taxonomy of this genus have classified them into two major clades, in which one of them comprises all pathogenic species, and the other, all saprophytic species. These clades are further subdivided into two subclades each. The pathogenic clade is subdivided into P1 and P2 subclades, which are also known as pathogenic and intermediate groups, respectively, and the saprophytic clade is subdivided into S1 and S2 subclades (Vincent et al, 2019).

*Leptospira* samples are also classified into more than 30 serogroups and 300 serovars (Brenner et al, 1999; Picardeau, 2017; Caimi & Ruybal, 2020). The serological classification is based on the structure of the lipopolysaccharides (LPS), which are complex molecules found in the outer layer of the outer membrane of some gram-negative bacteria. LPS is the main immunodominant antigens of these organisms, and differences in its sugar composition and

[1]Bioinformatics Multidisciplinary Environment (BioME), Instituto Metrópole Digital (IMD), Universidade Federal do Rio Grande do Norte, Natal, Brazil [2]Departamento de Engenharia de Computação e Automação (DCA), Centro de Tecnologia (CT), Universidade Federal do Rio Grande do Norte, Natal, Brazil [3]UMass Chain Medical School, Nonhuman Primates Reagent Resources, Department of Medicine, University of Massachusetts, Worcester, MA, USA

Correspondence: tetsu@imd.ufrn.br

orientation are the main characteristics that distinguish serovars (Adler, 2015). The genomic region responsible for LPS antigen biosynthesis is the *rfb* locus (Mitchison et al, 1997), and the genes in this region were demonstrated to be the main genetic factors associated with the serological classification of this genus (de la Peña-Moctezuma et al, 1999).

Serological identification is an essential procedure for epidemiology and for directing prophylactic measures, in the form of vaccines, to control the disease. However, the high serological diversity found in this genus imposes difficulties in developing more effective methods for serotyping and in designing broader and more comprehensive vaccines (Barazzone et al, 2022). The methods used to perform the serological classification, such as the microscopic agglutination test (Terpstra et al, 2003), are time-consuming and expensive. Molecular-based methods, such as PCR analysis and DNA sequencing, are promising alternatives to the current serotyping methods. Although there are suggestions in the literature, there are still no fully reliable molecular-based methods for performing this procedure (Sykes et al, 2022). This is due to the still scarce knowledge about the genetic basis associated with serological classification. Therefore, understanding how the serological diversity found in leptospires is generated in a genetic context could help improve or elaborate new strategies for disease prevention and control.

The number of genomes from the genus *Leptospira* has been significantly increasing in public databases over the years because of the increased processing capacity of sequencing technologies. All of these available data allow us to conduct studies that suggest genetic factors associated with the serological classification of *Leptospira spp.* strains. Therefore, this study aimed to further understand the genetic basis underlying the serological classification in *Leptospira*. For this, we accessed the genomic data of more than 700 samples of pathogenic species of *Leptospira* and performed a comparative analysis of the structure and organization of their *rfb* locus.

# Results

### Hierarchical clustering of *Leptospira* samples based on the genetic composition of the *rfb* locus

The *rfb* locus of the genus *Leptospira* comprises a broad variety of genes. To compare the gene composition of this locus among the samples used in this study, we performed a hierarchical clustering analysis using a presence/absence table of *rfb* locus genes. The table was created based on the ortholog groups (orthogroups) inferred from the proteomes of 722 pathogenic samples of *Leptospira* (Table S1). This generated a total of 10,084 orthogroups, of which 380 contained genes that are part of the *rfb* locus. After submitting the bi-dimensional table of the presence/absence of an orthogroup in each sample (Table S2) to the clustering method, a heatmap and 2-dimensional hierarchical clustering were generated (Figs 1 and S1).

In this analysis, samples were grouped into five well-defined clusters (clusters 1–5). Cluster stability analysis using the bootstrap approach also demonstrated high stability for all five clusters with a mean score of 99.14% and a minimum score of 98.1%. The distribution of serogroups among the clusters shows that samples belonging to one serogroup are generally in the same cluster (Table 1). Most of the samples from the srg Icterohaemorrhagiae (n = 155), srg Ballum (n = 71), and srg Sejroe (n = 61), which are the most frequent serogroups in the analysis, were grouped into clusters 2, 1, and 4, respectively. It is noteworthy that samples belonging to different species could belong to the same serogroups (Table 2). For instance, we could find five species among samples belonging to the srg Mini, *L. interrogans*, *Leptospira kirschneri*, *Leptospira borgpetersenii*, *Leptospira santarosai*, and *Leptospira mayottensis*, and all of them were in the same cluster. We also verified the distribution of the species in the five clusters (Table 3), and we did not find a clear pattern of their distribution among the clusters. The samples of species *L. interrogans* (n = 365), *L. borgpetersenii* (n = 160), and *L. santarosai* (n = 60), which are the most frequent species in the current study, can be found in five, three, and four clusters, respectively. All these observations indicate that the clustering generated in this work is independent of taxonomic classification and strongly associated with serological classification.

The analysis also involved genomic data from samples that did not have serological identification, which totalized 214 samples. Despite the lack of serological data, the inclusion of these samples is relevant to verify other organizations and structures of the *rfb* locus. All these samples were grouped into one of the five clusters previously mentioned, suggesting that they could share antigenic similarity with serologically characterized samples and could be classified into one of the existing serogroups. It is of high interest that serological identification of these samples is performed to confirm the findings of this study.

Hierarchical clustering by the orthogroups also allowed us to identify the orthogroups that are characteristic of each cluster (Figs 1 and S1). By evaluating the stability score of the cluster of orthogroups, referred to here as orthoclusters, using the bootstrap approach, we found high stability scores in almost all clusters when they were subdivided into nine orthoclusters with a mean and a minimum stability score of 91.52% and 76.72%, respectively. The orthocluster 4 is the largest orthocluster, and because we could depict distinct patterns of presence/absence among its orthogroups, we further subdivided it into five sub-orthoclusters (4a to 4e). By evaluating the cluster stability of the sub-orthoclusters, two of them (4a and 4d) presented very high stability (>85%), one (4c) presented high stability (75–85%), and two (4b and 4e) presented moderate stability (60–75%). In this analysis, it was possible to verify that orthoclusters 1, 2, and 3 contain orthogroups that characterize the samples from clusters 3, 1, and 4, respectively (highlighted in green in Fig 1). The sub-orthocluster 4e also presented orthogroups found only in, but not in all, samples of cluster 5.

### Comparative analysis of the structure and organization of the *Leptospira rfb* locus

To visualize the gene organization and to identify patterns among different profiles of the *rfb* locus in the genus *Leptospira*, we generated diagrams illustrating the genetic structure and organization of the *rfb* locus in representatives of each serogroup in the

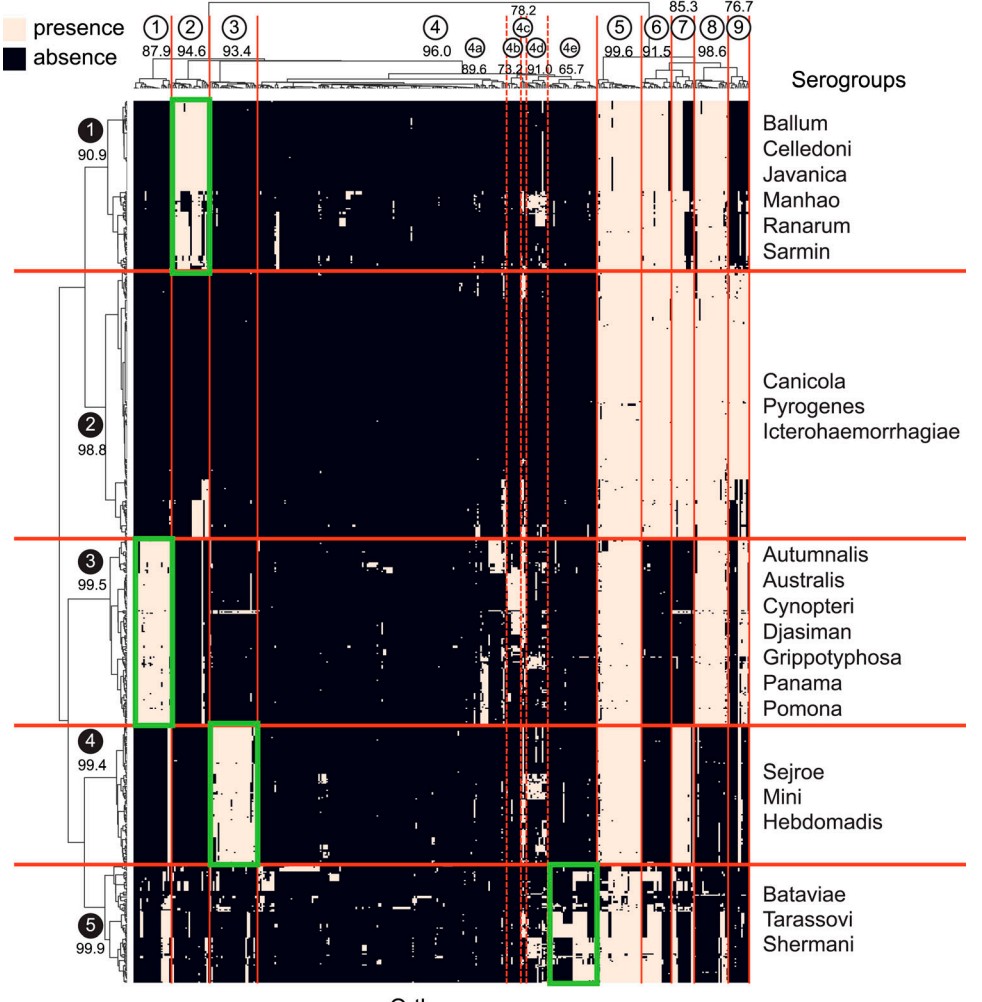

**Figure 1. Hierarchical clustering of pathogenic samples of *Leptospira* based on the gene composition of the *rfb* locus.**
Values in the heatmap are represented on a binary scale, with a darker color indicating zero (absence) and a lighter color indicating a value of 1 (presence). The tree on the x-axis represents the clustering of the orthogroups, whereas the tree on the y-axis represents the clustering of the samples. The samples were subdivided into five clusters, and orthogroups were subdivided into 9 (ortho)clusters. Orthocluster 4 was further subdivided into five sub-orthoclusters. Values near the cluster identification are stability scores calculated using the bootstrap approach. Areas delimited in green represent orthoclusters with orthogroups specific to a sample cluster.

analysis (Fig 2). In these diagrams, we could verify regions that are conserved in most samples and regions that vary considerably in the gene composition. The 3′-terminal region is one of the conserved regions. In addition to several genes that encode sugar-modifying enzymes, such as methyltransferases and glycosyltransferases, this region comprises genes responsible for the O-antigen processing and synthesis through the Wzy-dependent pathway, and for the dTDP-rhamnose biosynthesis (*rfbC, rfbD, rfbB, rfbA*), which are involved in the assembly of LPS. In the 5′ region of the *rfb* locus, we can find in all representative samples the genes *MarR*, which encodes for a transcriptional regulator that initiates the *rfb* locus, and the GDP-L-fucose synthase (*fcl*). Following the *fcl*, we could also verify a set of genes encoding sugar-modifying enzymes conserved in samples from clusters 1, 2, and 3.

The innermost region of the *rfb* locus has a more diverse gene composition among the analyzed samples. This region consists mainly of genes encoding enzymes that add or modify sugar during the O-antigen biosynthesis. Comparing the gene composition of the locus between samples from different clusters, we can see that each cluster presents a distinctive gene composition, except for the

samples from clusters 1 and 2. For this reason, we decided to join these two clusters and named the new group class I. Clusters 3, 4, and 5 were renamed respectively class II, III, and IV.

Interestingly, we could observe several small syntenic blocks in samples from different species, classes, and serogroups (colored bars in Fig 2). The different combinations of these syntenic blocks at the *rfb* locus suggest that they are determinants in expressing the serogroup of a sample and are promising molecular markers for serogroup identification. Taking the sample of srg Icterohaemorrhagiae (class I), it can be seen that there is a set of genes (highlighted with a brown bar, Fig 2) that can differentiate samples of this serogroup from srg Canicola and srg Pyrogenes. But because this same set of genes can be found in other samples, such as of the srg Javanica (class I), srg Autumnalis, srg Cynopteri, and srg Panama (class II), a second set of markers is needed. This may be a gene from a set that is found only in samples from srg Icterohaemorrhagiae and srg Canicola (highlighted with light blue bar, Fig 2). The specificity of these gene sets in srg Icterohaemorrhagiae was confirmed by verifying their conservation in almost all samples classified as srg Icterohaemorrhagiae (152/155 samples) and their

**Table 1. Distribution of serogroups in the analyzed samples, highlighting the clustering of samples belonging to the same serogroup.**

| Serogroups | Clusters | | | | | Total |
|---|---|---|---|---|---|---|
| | 1 | 2 | 3 | 4 | 5 | |
| Ballum | 71 | | | | | 71 |
| Javanica | 15 | | | | | 15 |
| Celledoni | 4 | | | | | 4 |
| Manhao | 3 | | | | | 3 |
| Sarmin | 3 | | | | | 3 |
| Ranarum | 2 | | | | | 2 |
| Icterohaemorrhagiae | 2 | 152 | 1 | | | 155 |
| Pyrogenes | 7 | 27 | 1 | | | 35 |
| Canicola | 1 | 11 | | | 1 | 13 |
| Grippotyphosa | | | 32 | | 1 | 33 |
| Pomona | | | 25 | | | 25 |
| Autumnalis | 2 | 22 | | | | 24 |
| Australis | | 9 | | | | 9 |
| Cynopteri | | 1 | | | | 1 |
| Djasiman | | 1 | | | | 1 |
| Panama | | 1 | | | | 1 |
| Sejroe | | | 1 | 60 | | 61 |
| Mini | | | | 9 | | 9 |
| Hebdomadis | | | | 6 | | 6 |
| Tarassovi | | | | | 18 | 18 |
| Bataviae | | | | 1 | 14 | 15 |
| Shermani | | | | | 4 | 4 |
| Unknown | 29 | 31 | 57 | 37 | 59 | 214 |
| Total | 139 | 221 | 151 | 114 | 97 | 722 |

absence in samples from other serogroups (Fig 3 and Table S3). The occurrence of these syntenic blocks in samples of the same species but different serogroups also suggests that this region is prone to lateral transfer events. Phylogenetic analysis is of great interest to understand the events of change or the emergence of new serogroups.

Although the genes involved in the biosynthesis of the O-antigen usually cluster at the *rfb* locus, some samples presented segments of this locus in other genome regions. Among them is the sample *L. interrogans* srg Pomona srv Pomona str AKRFB (Fig 2). In this sample, we could notice a long segment containing genes homologous to the internal genes of the *rfb* locus of other samples, but located outside the locus.

## Functional diversity of the *Leptospira rfb* locus

To analyze the functional diversity of the proteins encoded by the *rfb* locus, we extracted the amino acid sequences of these proteins from the 22 representative samples and submitted them to the InterProScan program (Jones et al, 2014). The InterProScan performs protein annotation using up to 11 applications. Among them, the SUPERFAMILY database (Pandurangan et al, 2019) was one of those that annotated the most proteins in the fewest functional categories. Therefore, we restricted the analysis to domain signatures from the SUPERFAMILY database.

A total of 1,714 proteins were extracted, and of these, 1,366 proteins (79.69%) showed a match with at least one SCOP superfamily (Table S4). The representative samples had an average of 80.39% of their proteins matched with at least one SCOP superfamily. The samples with the lowest and highest proportions were respectively from the srg Sarmin (GCF_030023765.1), with 58.88%, and the srg Djasiman (GCF_000216195.1), with 85.53% (Fig 4A and Table S4). Most of these proteins (1,146 proteins) had a match with only one distinct SCOP superfamily. The other proteins had matches with two (193 proteins), three (17 proteins), or four (9 proteins) distinct SCOP superfamilies. The number of distinct SCOP superfamilies found among the proteins of the *rfb* locus totaled 61. Of these, only 11 were found in all 22 samples and 15 were found in at least 75% (17) of the samples (Fig 4B and Table S4), indicating that most of the functional groups found must perform accessory functions that contribute to the structural diversity of the LPS found in this genus. It is also noteworthy that some SCOP superfamilies are abundant and can be found in multiple proteins in the same sample (Fig 4C). Among the five most abundant superfamilies, nucleotide-diphospho-sugar transferases (318 proteins), NAD(P)-binding Rossmann-fold domains (252 proteins), S-adenosyl-L-methionine–dependent methyltransferases (156 proteins), PLP-dependent transferases (111 proteins), and RmlC-like cupins (79 proteins) were found respectively in 12–20, 8–18, 1–12, 2–9, and 2–9 proteins per sample (Table S4). These five SCOP superfamilies characterize proteins with enzymatic activity that can modify the structure of LPS. By grouping the samples into their respective classes, class I showed a higher average number of proteins in all five SCOP superfamilies (Fig 4D).

Finally, we applied a hierarchical clustering algorithm using the presence/absence table of the 61 superfamilies among the 22 representative samples (Figs 5 and S2 and Table S5) to contemplate the distribution of the functional groups among the samples and classes. In this analysis, samples belonging to the same class were grouped into the same cluster with a very high stability score (>85%), indicating that they also share similar functional profiles. By verifying the clusters composed of SCOP superfamilies, we could also verify which functional groups are common to all or characterize each of the four sample classes (Figs 5 and S2). We grouped the SCOP superfamilies into nine clusters (SP1 to SP9, Fig 5), which returned clusters with mean and minimum stability scores of 63.0% and 77.9%, respectively. The cluster SP2 is comprised of SCOP superfamilies that are found in most of the representative samples and represents those functional groups that are essential for the LPS biosynthesis, such as "Winged helix" DNA-binding domain (found in the MarR) and ABC transporter transmembrane region, and for LPS modification, such as S-adenosyl-L-methionine–dependent methyltransferases. Meanwhile, the cluster SP6 is comprised of SCOP superfamilies that are specific to a few samples. The other clusters have a distinguished distribution along the samples and can be used to characterize the functional profile of each sample class. For instance, we could establish that typical

**Table 2. Distribution of serogroups among some pathogenic species of *Leptospira*.**

| Serogroups/species | Lint | Lbor | Lsan | Lkir | Lnog | Lwei | Lyas | Lkme | Lale | Lmay |
|---|---|---|---|---|---|---|---|---|---|---|
| Ballum | | 71 | | | | | | | | |
| Javanica | | 12 | 3 | | | | | | | |
| Celledoni | | | | | | 4 | | | | |
| Manhao | | | | | | 1 | | | 2 | |
| Sarmin | | | 2 | | | | | | 1 | |
| Ranarum | | | | | | 1 | | | | |
| Icterohaemorrhagiae | 153 | | | 1 | | 1 | | | | |
| Pyrogenes | 28 | | 6 | | 1 | | | | | |
| Canicola | 13 | | | | | | | | | |
| Grippotyphosa | 12 | | 7 | 14 | | | | | | |
| Pomona | 18 | | | 7 | | | | | | |
| Autumnalis | 15 | | | 3 | 4 | | | | 2 | |
| Australis | 7 | | | | 2 | | | | | |
| Cynopteri | | | | 1 | | | | | | |
| Djasiman | 1 | | | | | | | | | |
| Panama | | | | | 1 | | | | | |
| Sejroe | 15 | 40 | 6 | | | | | | | |
| Mini | 1 | 1 | 5 | 1 | | | | | | 1 |
| Hebdomadis | 3 | | 2 | | | 1 | | | | |
| Tarassovi | | 8 | 8 | | | | | 1 | 1 | |
| Bataviae | 13 | | | | 2 | | | | | |
| Shermani | | | 4 | | | | | | | |
| Unknown | 86 | 28 | 17 | 11 | 16 | 13 | 8 | 6 | | 5 |

Lint, *L. interrogans*; Lbor, *L. borgpetersenii*; Lsan, *L. santarosai*; Lkir, *L. kirschneri*; Lnog, *L. noguchii*; Lwei, *L. weilii*; Lyas, *L. yasudae*; Lkme, *L. kmetyi*; Lale, *L. alexanderi*; Lmay, *L. mayottensis*.

functional groups found among the proteins encoded by the locus *rfb* of samples of class I are those grouped into the clusters SP1, SP3, and SP9. Meanwhile, for samples of class II, functional groups grouped into clusters SP1 and SP5 could be considered as typical for this class. All these results demonstrate that the diversity in the type, quantity, and combination of functional groups in the locus *rfb* along the samples is intimately associated with the LPS structural diversity found in this genus.

## Discussion

The association between the *rfb* locus and the serological classification in *Leptospira* was first verified by comparing this locus among samples of *L. interrogans* serovar Hardjo, *L. borgpetersenii* serovar Hardjobovis, and *L. interrogans* serovar Copenhageni (de la Peña-Moctezuma et al, 1999, 2001). In these studies, it was observed that the genes of samples belonging to the same serological class but not to the same species shared high similarity when comparing the genes comprising this locus. Subsequent studies also showed evidence of this association, such as the similarity in the gene composition of the *rfb* locus between samples of the Hurstbridge

serovar of the *Leptospira fainei* and *Leptospira broomi* species (Fouts et al, 2016) and among samples of the Sejroe, Hebdomadis, and Mini serogroups, which share antigenic affinity (Medeiros et al, 2022).

A study involving the main serogroups also verified the similarity of the gene profile of the *rfb* locus among samples classified in the same serogroup (Nieves et al, 2022). In this work, we were able to further investigate the diversity of gene profiles in the *rfb* locus and the patterns that characterize a serogroup by accessing the genomic data of 722 pathogenic *Leptospira* samples distributed across 20 species and 22 serogroups. The genetic structure of the *rfb* locus observed among the samples analyzed in this study followed as described in the previous study (Fouts et al, 2016). In general, the *rfb* locus begins with the *MarR* regulatory gene and ends with the *DASS* gene. The genes located at the 3'-terminal of the locus are conserved and consist of genes responsible for the O-antigen synthesis and LPS assembly. Meanwhile, the composition of the genes at the 5'-terminal is more variable and consists mainly of sugar-modifying enzymes. In some cases, we observed in some samples long segments containing genes homologous to the internal genes of the *rfb* locus located outside the locus. One could suggest that we are dealing with samples that have low sequencing

**Table 3. Distribution of *Leptospira* species in each cluster.**

| Species | Clusters | | | | | Total |
|---|---|---|---|---|---|---|
| | 1 | 2 | 3 | 4 | 5 | |
| *L. adleri* | | | | | 3 | 3 |
| *L. ainazelensis* | | | | | 1 | 1 |
| *L. ainlahdjerensis* | | | | | 1 | 1 |
| *L. alexanderi* | 5 | | | | 1 | 6 |
| *L. alstonii* | 4 | | 1 | | | 5 |
| *L. barantonii* | | | | | 2 | 2 |
| *L. borgpetersenii* | 99 | | | 47 | 14 | 160 |
| *L. ellisii* | | | | | 2 | 2 |
| *L. gomenensis* | | | | | 4 | 4 |
| *L. interrogans* | 4 | 215 | 96 | 33 | 17 | 365 |
| *L. kirschneri* | | 4 | 29 | 4 | 1 | 38 |
| *L. kmetyi* | | | | | 7 | 7 |
| *L. mayottensis* | | 2 | | 3 | 1 | 6 |
| *L. noguchii* | 2 | | 15 | | 9 | 26 |
| *L. sanjuanensis* | | | | | 2 | 2 |
| *L. santarosai* | 12 | | 10 | 20 | 18 | 60 |
| *L. stimsonii* | | | | | 4 | 4 |
| *L. tipperaryensis* | | | | | 1 | 1 |
| *L. weilii* | 13 | | | 7 | 1 | 21 |
| *L. yasudae* | | | | | 8 | 8 |
| Total | 139 | 221 | 151 | 114 | 97 | 722 |

corresponding to class I, whereas in this work, 25 samples of srg Pomona clustered in class II.

Comparative analysis of the gene profile of the *rfb* locus also allowed us to observe the existence of some small syntenic blocks differently distributed between samples and the existence of a unique combination of these blocks that characterize samples of a serogroup. This was analyzed in detail for the samples of srg Icterohaemorrhagiae in this work. A similar observation was also made previously between samples from the Sejroe, Hebdomadis, and Mini serogroups (class III) (Medeiros et al, 2022), which characterized a small region of the *rfb* locus comprised of 3–4 genes whose gene composition varies according to their serological classification. More detailed studies for samples from other serogroups are needed, but this is strong support that it is possible to establish a reasonable number of molecular markers to enable serological identification procedure using molecular-based methods as remarked in other studies (Medeiros et al, 2022; Nieves et al, 2022).

Although most samples of a serogroup were clustered into one of the four classes established in this study, eight samples did not follow this pattern. In the analysis made with the representative sample of srg Icterohaemorrhagiae, we also found three of 155 samples classified as srg Icterohaemorrhagiae without one of the syntenic blocks that characterize this serogroup. There are a few reasons that could contribute to this. One of them is the use of genomes with poor assembly quality, which can affect the annotation of their genes and, consequently, the clustering analysis. Although they showed high completeness scores in BUSCO (>98.3%), the *rfb* locus was fragmented in some samples, which could affect their clustering. Another reason could be the concomitant occurrence of genes that are characteristic of two or more classes in a sample. This is the case for sample *L. interrogans* str UT053, which is classified as from srg Sejroe (srv Medanensis), but clustered in class II instead of class III. When checking the orthogroups found in this sample, we noticed that it has orthogroups from both orthocluster 1, characteristic of class II, and orthocluster 3, characteristic of class III. Another possibility could be the acquisition of a small number of genes that affect the serology of a sample because differences in a few genes could be enough to distinguish samples of different serogroups (Medeiros et al, 2022; Nieves et al, 2022; Chinchilla et al, 2023). Finally, another reason that should not be ruled out is the misannotation of the serological data. This could happen because of accidental contamination or difficulties in interpreting the microscopic agglutination test results (Goris & Hartskeerl, 2014). Unfortunately, we could not access serological test results for these samples to verify this. In this sense, revisions in the serological classification of some samples are of interest to confirm our findings. For instance, the *rfb* locus profile of samples *Leptospira alexanderi* srv Erinaceiauriti str 56159 and *L. alexanderi* srv Nanla str 56650 (Xu et al, 2016), both currently classified in the srg Autumnalis, has a high similarity to the profiles of samples from the srg Manhao (class I), suggesting that the most appropriate classification for them is the srg Manhao.

Because we sought to emphasize the pathogenic group, the present study did not analyze the genetic composition of the *rfb* locus in other leptospiral groups (intermediate and saprophytic). A detailed analysis of the *rfb* locus in these groups is in our perspective work because it could bring important insights into the evolutionary dynamics of serology in *Leptospira*.

quality and, consequently, with assembly errors. However, there are samples with well-assembled genomes that also showed this characteristic. This observation should be considered when attempting to identify the serology of samples using genomic data.

Here, we also propose the clustering of the samples into four classes (class I, II, III, and IV) according to their gene composition of the *rfb* locus. In this clustering, we could observe that it is consistent with the serological classification because most of the samples belonging to a serogroup are grouped into the same class. This also suggests that samples belonging to the same class might share antigenic affinity. This is supported by observing the clustering of some samples from serogroups that were classified indistinctly in the past. The serogroups Sejroe, Mini, and Hebdomadis, which comprise class III, were once part of a single serogroup called Hebdomadis, which was separated in 1982 (Stallman, 1984). Similarly, the serogroups Autumnalis and Djasiman were part of a single serogroup called Autumnalis (Stallman, 1984) and they are clustered in class II. Clustering analysis based on gene composition of the *rfb* locus was also performed concomitantly with this work and recently published by Chinchilla et al (2023). Although they carried out the analysis with a smaller number of samples, they also could group the samples into four clusters, which correspond to the four classes proposed in this work. The only difference was the placement of the serogroup Pomona. In Chinchilla et al (2023), a single sample of srg Pomona was placed in the cluster

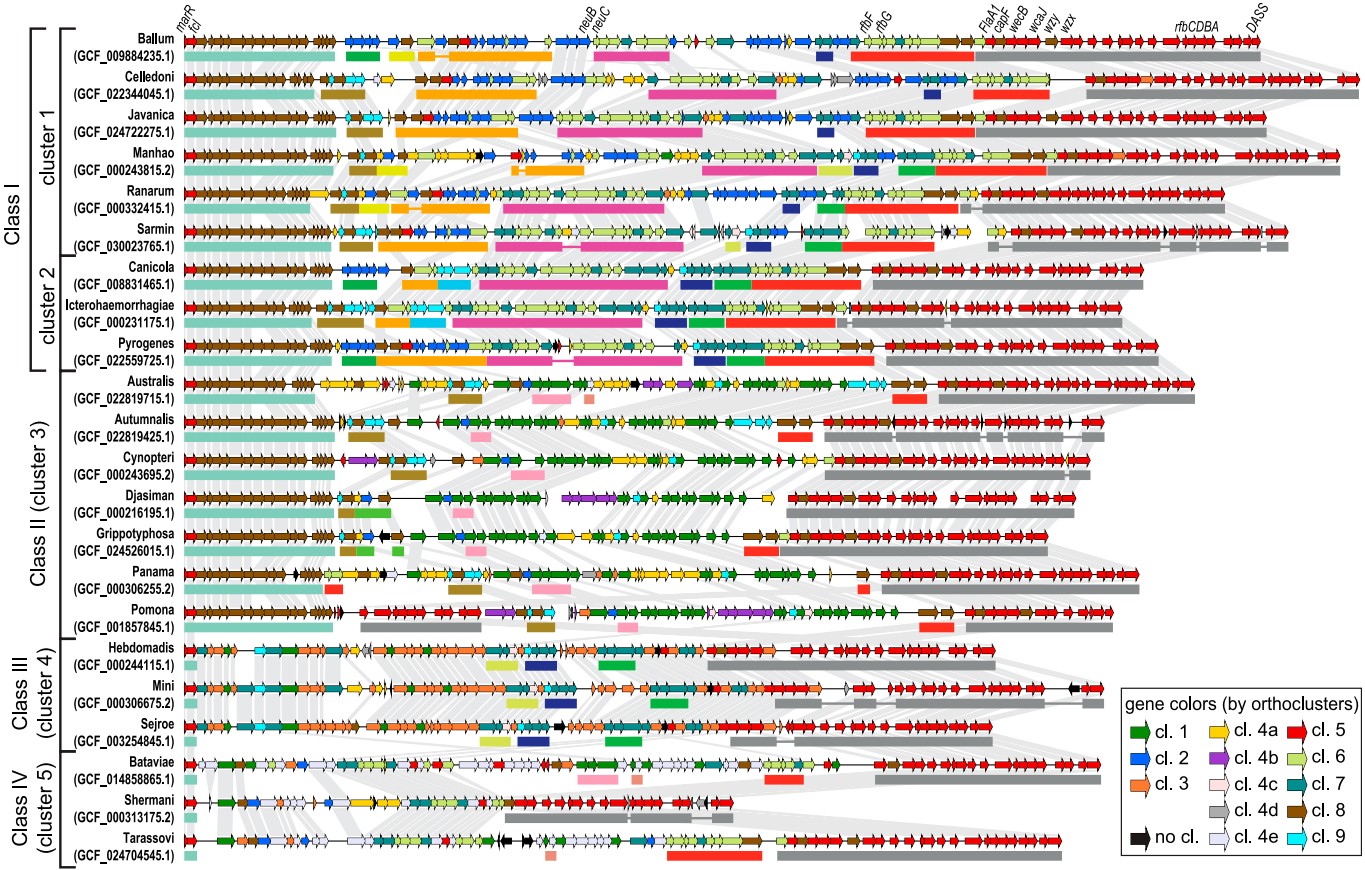

**Figure 2. Genetic organization of the *rfb* locus in representative samples of each serogroup.**
Genes that have the same color belong to the same orthocluster. Genes in black belong to orthogroups that were not included in the clustering analysis (see the Materials and Methods section). Gene pairs from adjacent samples that are from the same orthogroup and reciprocal best BLAST hits are linked. Colored bars below some gene blocks highlight syntenic blocks shared among the samples (bar colors do not correspond to the gene colors). Syntenic blocks involving only samples of the same cluster were omitted for clarity, except for those involving the sample of srg Icterohaemorrhagiae. The gene symbols of some genes were placed at the top for reference.

The significant differences in antigen biosynthetic enzymes and the structural organization of the leptospire *rfb* locus reveal an important field to be explored in this bacterial genus. Additional studies, such as knockout genes, are needed to evaluate the antigenic relationship between different serogroups and to determine whether the observed genetic similarity reflects significant cross-reactivity in the host immune response. Regardless, the identification of genetic and functional similarities in the *rfb* locus between samples of different serogroups shown in this study can guide the development of broader and more effective diagnostic techniques and vaccines against leptospirosis. Ultimately, we highlight the importance of ongoing efforts to improve the quality of *Leptospira* genomes to improve our understanding of this genus.

# Materials and Methods

### Sources of genomic and serological data

The dataset used for this study is based on the dataset used in Medeiros et al (2022) with some updates in the serological classification of some samples. The dataset consists of proteome and serological data from 722 samples of pathogenic leptospire, which were obtained respectively from the National Center for Biotechnology Information (NCBI) Reference Sequence Database (RefSeq) and the Bacterial Isolate Genome Sequence Database (Jolley & Maiden, 2010) (http://bigsdb.pasteur.fr/). The data were obtained in December 2023. In addition, serological data of some samples were obtained from the literature and incorporated into the dataset. The samples studied in this work encompass 20 distinct pathogenic species (clade P1). Of the 722 samples used in the study, 214 samples did not have a serogroup determined. The remaining samples were classified into one of 22 serogroups (Table S1).

We also used the Benchmarking Universal Single-Copy Orthologs (BUSCO) with the spirochaetia_odb10 dataset to measure genome assembly and annotation completeness (Simão et al, 2015). The minimum and maximum completeness scores found among the samples were 71.1% and 100%, respectively, in which more than 99.3% of samples showed a score above 95% (Table S1). Despite the existence of a few genomes (five samples) with low quality, they were not initially discarded because the focus of this study was not to analyze the genome as a whole but rather the *rfb* locus.

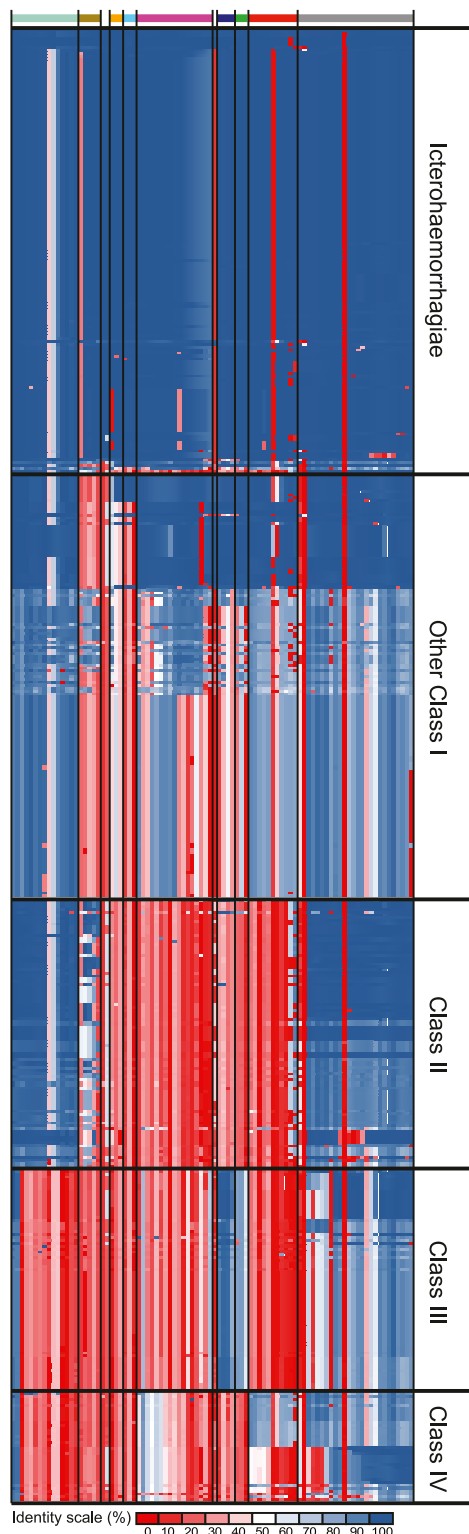

**Figure 3. Blast sequence similarity analysis of 88 proteins (columns) encoded by the *rfb* locus of the representative sample of srg Icterohaemorrhagiae (*L. interrogans* srv Lai str IPAV, GCF_000231175.1) with other *Leptospira* strains (lines).**
Colored bars on the top correspond to the syntenic blocks highlighted in Fig 2. Genes encompassed by the brown and light blue bars are concomitantly found in almost all samples of srg Icterohaemorrhagiae.

## Ortholog cluster inference and determination of *rfb* locus gene orthogroups

We submitted the entire set of protein sequences of all 722 samples to the OrthoFinder 2 (Emms & Kelly, 2019) software tool (with the parameter -y) to determine the orthologous groups between the gene sequence sets in the different accesses. The program was executed firstly until the inference of a species tree. Then, we inspected the species tree and rooted it by the clade composed of samples from species *Leptospira ellisii* and *Leptospira gomenensis*, which diverged earliest in the P1 clade (Vincent et al, 2019). The program generated 10,084 orthologous groups, where 404 were present in single copy in all samples, and 506 were present in all samples, but not necessarily in single copy.

We subsequently identified the orthogroups that compose the *rfb* locus. For this, we selected samples that contained the *rfb* locus assembled in a single contig. This was checked by verifying whether the genes that initiate (*MarR*) and terminate (*DASS*) the locus were contained within the same contig. Then, we identified the orthogroups of the genes located between *MarR* and *DASS*. These procedures were made using an in-house Python script. We found 314 samples with the intact *rfb* locus from 20 of 22 serogroups. Serogroups that were not represented among the 314 samples (Sarmin and Djasiman) had the genes belonging to the *rfb* locus extracted through a manual inspection of their genomic data. Of the total of 10,084 orthogroups, 422 contained genes that are part of the *rfb* locus. Among the 422 orthogroups, we observed the occurrence of some orthogroups that were mostly composed of proteins located outside the *rfb* locus and that may not be part of this locus. Thus, we removed those orthogroups in which the proteins were in the *rfb* locus in a few samples (<5) and outside the *rfb* locus in several samples (>50). In the end, 380 orthogroups were selected.

Representative samples from each serogroup were selected to conduct the visual analysis of the gene structure of the *rfb* locus (Fig 2) and to perform the functional analysis of its genes. Representative samples were selected in such a way as to prioritize those samples that have the *rfb* locus assembled in a single contig, better genome assembly quality, and, finally, a greater number of genes in the locus.

## Functional annotation of proteins encoded by the locus *rfb*

To functionally annotate the proteins encoded by the locus *rfb*, we extracted their amino acid sequences and submitted them to the InterProScan (Jones et al, 2014) available on the InterPro web server (https://www.ebi.ac.uk/interpro/search/sequence/). In this analysis, we restricted the search against domain signatures of the SUPER-FAMILY database (Pandurangan et al, 2019), which has a collection of hidden Markov models representing each protein structural domain at the superfamily level of the SCOP database. An E-value threshold of $1 \times 10^{-5}$ was used to filter the results. Sequence manipulation in this step was made using the software SeqKit (Shen et al, 2016).

## Clustering, visualization, and similarity of the genetic/functional composition of the *rfb* locus

The hierarchical clustering method was used to group samples/ orthogroups or samples/functional groups with similar genetic/

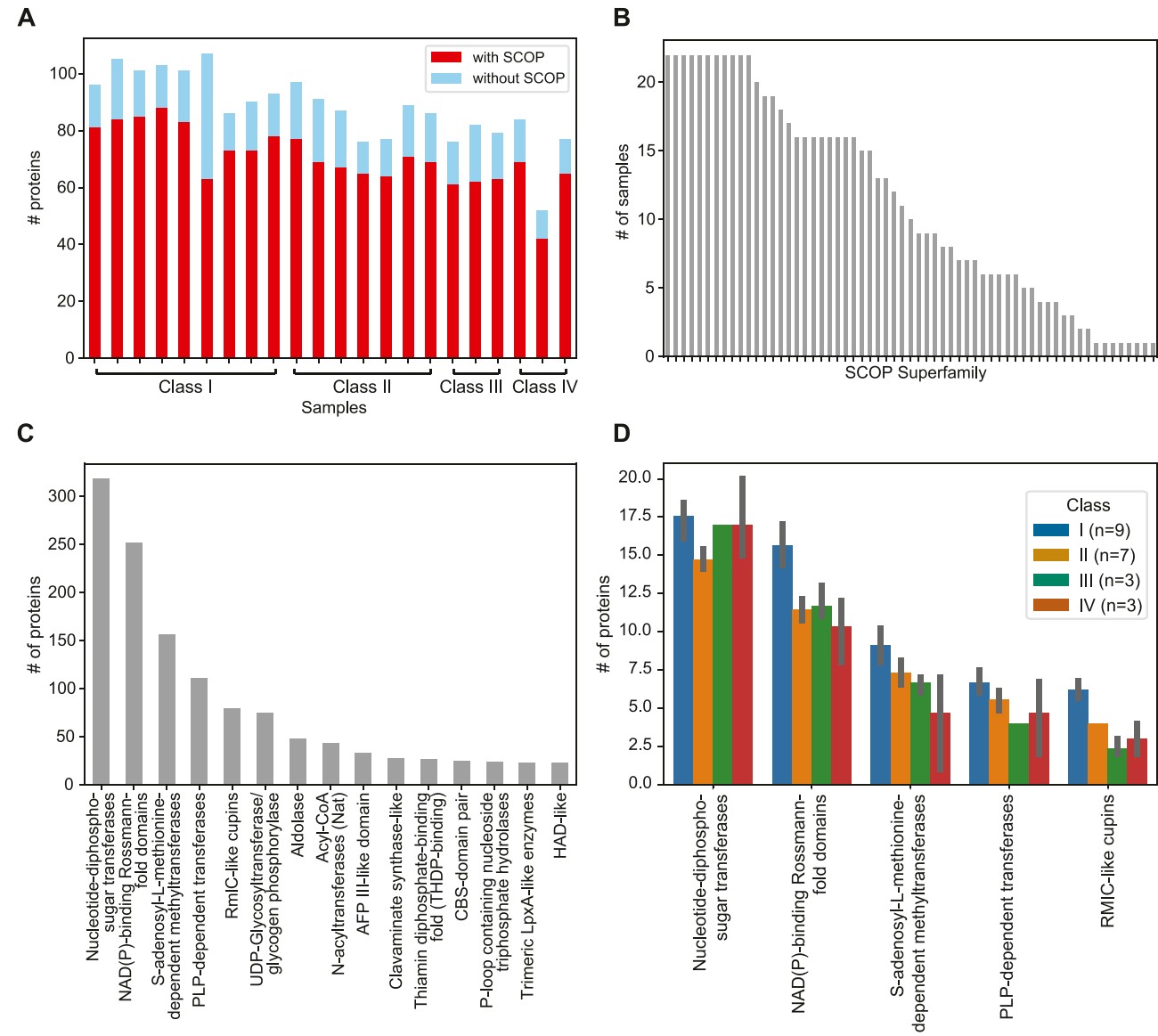

**Figure 4. Functional analysis of proteins encoded in the *rfb* locus among 22 representative samples of *Leptospira*.**
**(A)** Proportion of proteins assigned to at least one SCOP superfamily for each sample (red). The height of the bars corresponds to the number of proteins encoded by the *rfb* locus. **(B)** Distribution of the number of samples that present a SCOP superfamily. **(C)** Top 15 most abundant SCOP superfamilies. **(D)** Average number of the five most abundant SCOP superfamilies among the samples grouped by class. Error bars indicate a confidence interval of 95%.

functional profiles in the *rfb* locus. This analysis was performed using the scikit-learn (Pedregosa et al, 2011), pandas (McKinney, 2010), SciPy (Virtanen et al, 2020), and seaborn (Waskom, 2021) Python libraries. The Ward method was used for clustering, and the Euclidean distance metric was used for measuring the distance between samples, orthogroups, or functional groups. To assess the stability of the clusters, we used the function clusterboot (Hennig, 2007) implemented in the R library fpc. This method uses the bootstrap approach, which calculates the Jaccard similarity between the original clusters and the most similar cluster in the resampled data. All analysis was conducted with 1,000 replicates. Support values above 0.85, between 0.7 and 0.85, between 0.6 and

0.7, and below 0.6 were considered as very high, high, moderate, and low stability, respectively. To decide on an appropriate number of clusters (k), we calculated the cluster stability for a range of values of k (4–10) and chose the one in which the minimum and mean stability scores among the clusters showed higher values.

To generate the diagram of the genetic composition of the *rfb* locus in the representative samples (Fig 2), we used the command-line software Linear Display (https://github.com/JCVenterInstitute/ LinearDisplay). Two proteins from different samples were linked if they were in the same orthogroup and if they were reciprocal best BLAST hits. Syntenic blocks were determined with the aid of the software CSBFinder (Svetlitsky et al, 2019) with the following

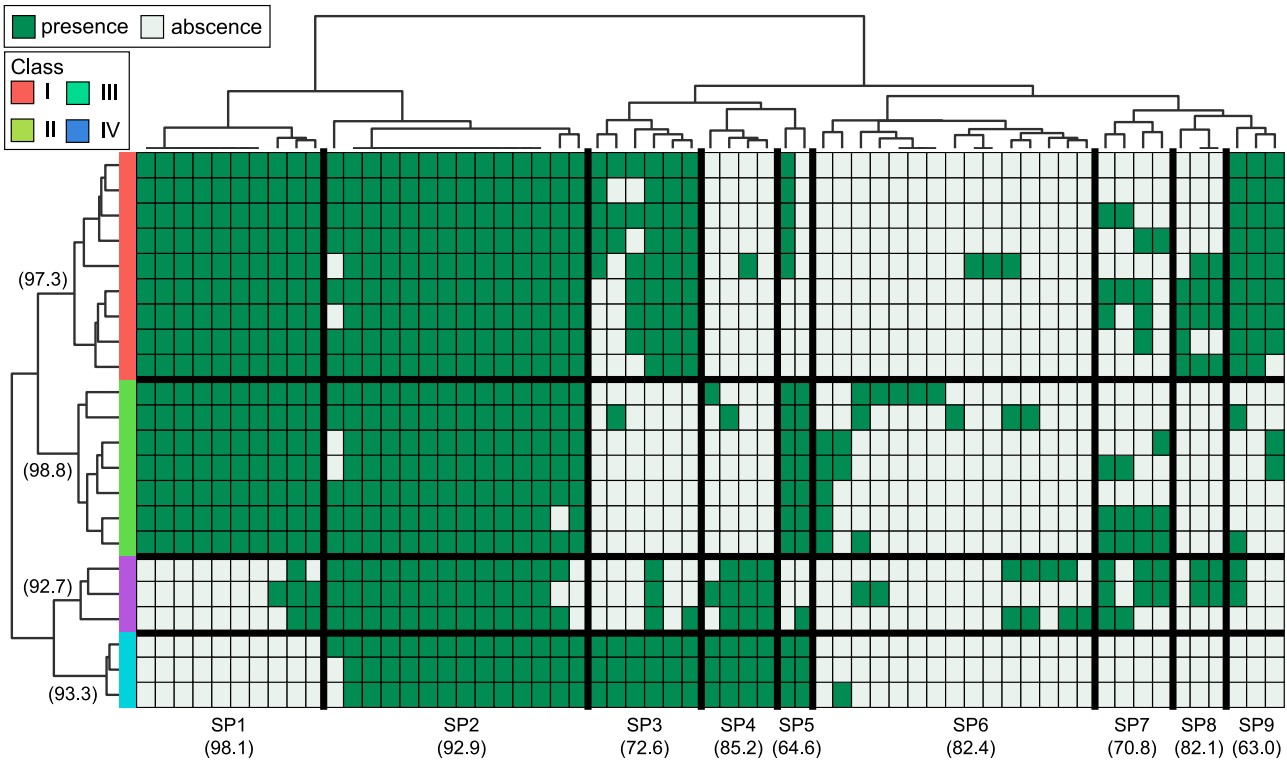

**Figure 5. Hierarchical clustering of 22 *Leptospira* representative samples based on the functional groups found among proteins encoded in the *rfb* locus.**
Values are represented on a binary scale, with a light gray color indicating 0 (absence) and a green color indicating 1 (presence). The tree on the x-axis illustrates the clustering of the functional groups (SCOP superfamilies), whereas the tree on the y-axis, the clustering of the samples. The functional groups were grouped into nine clusters (SP1 to SP9). Values in parentheses are the stability scores of each cluster.

parameters: quorum: 3; number of insertions: 0; and minimum length: 2.

The conservedness between the two genes shown in Fig 3 was accessed by sequence similarity search using the BLAST algorithm (blastp) (Altschul et al, 1990). The identity score between the query and subject proteins was calculated as the product of the percentage of identity and the query coverage of the first high-scoring segment pair as performed by Medeiros et al (2022).

## Data Availability

The sequence data that support the findings of this study are openly available in NCBI Genome at https://www.ncbi.nlm.nih.gov/genome/. All sequence accessions of the samples used in this work are listed in its supplementary material.

## Supplementary Information

## Acknowledgements

This study was financed in part by Coordenação de Aperfeiçoamento de Pessoal de Nível Superior—Brazil (CAPES)—Finance Code 001. We would like to thank the teams from Bioinformatics Multidisciplinary Environment (BioME/IMD) at UFRN, Centro de Processamento de Alto Desempenho (CEPAD/ICB) at UFMG, and High-Performance Computing Center (NPAD) at UFRN for the support on the computational resource. We are also grateful to the Institut Pasteur teams for the curation and maintenance of BIGSdb-Pasteur databases at http://bigsdb.pasteur.fr/.

### Author Contributions

L CA Ferreira: software, formal analysis, investigation, visualization, and writing—original draft.
L de FA Ferreira Filho: software and investigation.
MR V Cosate: conceptualization and writing—review and editing.
T Sakamoto: conceptualization, supervision, validation, methodology, and writing—review and editing.

### Conflict of Interest Statement

The authors declare that they have no conflict of interest.

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
