## [Reviewer comments · Life Science Alliance]

Life Science Alliance

Genetic structure and diversity of rfb locus of pathogenic species of genus *Leptospira*

Leonardo Cabral Afonso Ferreira, Luiz de França Afonso Ferreira Filho, Maria Venturim Cosate, and Tetsu Sakamoto

DOI: <https://doi.org/10.26508/lsa.202302478>

Corresponding author(s): Tetsu Sakamoto, Federal University of Rio Grande do Norte

Review Timeline:

Submission Date:	2023-11-10
Editorial Decision:	2023-12-12
Revision Received:	2024-02-18
Editorial Decision:	2024-03-07
Revision Received:	2024-03-08
Accepted:	2024-03-08

Transaction Report:

December 12, 2023

Re: Life Science Alliance manuscript #LSA-2023-02478-T

Tetsu Sakamoto
Universidade Federal do Rio Grande do Norte

Dear Dr. Sakamoto,

Thank you for submitting your manuscript entitled "Genetic structure and diversity of rfb locus of pathogenic species of genus *Leptospira*" to Life Science Alliance. The manuscript was assessed by expert reviewers, whose comments are appended to this letter. We invite you to submit a revised manuscript addressing the Reviewer comments.

Thank you for this interesting contribution to Life Science Alliance. We are looking forward to receiving your revised manuscript.

Sincerely,

B. MANUSCRIPT ORGANIZATION AND FORMATTING:

Reviewer #1 (Comments to the Authors (Required)):

Genetic structure and diversity of the rfb locus of pathogenic species of genus *Leptospira*
By Ferreira et al.

The study conducted by Ferreira and colleagues aimed to comprehensively characterize the rfb locus of *Leptospira*, which encodes enzymes involved in O-antigen biosynthesis, from both a genetic and functional perspective. Utilizing a dataset comprising 722 genomes, the researchers performed a comparative analysis of gene composition within the rfb locus, resulting in the classification of samples into five distinct classes. Each class encompassed various serogroups sharing serological affinity.

The first part involved defining the distribution of orthologous genes from the rfb across the identified classes, revealing specific "orthogroups" that distinguished serogroups. Subsequently, the researchers assessed synteny and gene composition within the rfb across the classes, proposing syntenic blocks as potential molecular markers. The study concluded with a functional characterization of proteins within superfamilies, providing insights into the representation of certain protein families across different classes. If the first part has already been published by others, the functional characterization is of particular interest, as it offers valuable insights into specific modifications of LPS shared by groups of serogroups, distinguishing them from others. Overall, the article is well-written and contributes significantly to the field, although certain aspects may benefit from additional explanation.

Comments:

- Statistics: One of the major comments is the lack of statistics to support the division in clusters or orthogroups.
- Methodology: a lot of details are missing see below.
- Recently, Chinchilla et al, PLOS NTD (10.1371/journal.pntd.0011733), proposed 4 clusters. Although they used less genomes how it compares to yours?
- Page 4: "Recent studies... have identified 67 species so far and classified them into two major clades". There is no reference associated to this sentence, at present, there are 69 species, considering the inclusion of *L. sanjuanensis*, reported in Fernandes et al., 2022.

In the initial part, 395 orthologous groups associated with the rfb were identified. Upon reviewing the methodology, the general approach employed to delineate these orthologous groups is described, utilizing as reference those genomes with an intact rfb locus, where marR was positioned at the 5' end and DASS at the 3' end. However, several details, such as the specific software employed and the associated parameters, are not explicitly disclosed and should be incorporated for comprehensive clarity.

Figure 1 lacks clarity regarding the methodology employed for clustering orthologous genes. Although the text mentions the definition of a threshold, this is not included in the methodology section. Additional details would enhance the clarity of the analysis. In Figure 2, the orthoclusters are more evident, with distinct groups distinguished by coloration. Including a broad functional characterization, if feasible, for these orthoclusters might improve general comprehension.

"Most of the samples from the srg *Icterohaemorrhagiae* (n=117), srg *Ballum* (n=66), and srg *Sejroe* (n=46), which are the most frequent serogroups in the analysis, were grouped into clusters 1, 2, and 4, respectively" - *Icterohaemorrhagiae* is in cluster 2, *Ballum* in 1 and *Sejroe* in 4. The text should be modified.

In Table 1, it is apparent that, despite most serogroups are predominantly clustering within a single group, certain genomes from specific serogroups exhibit clustering in more than one group. For example, *Autumnalis*: 2 genomes are in cluster 1 and, other 9 genomes in cluster 3. Is it attributed to genome fragmentation? The discussion surrounding genome fragmentation is entirely absent, and the researchers do not extensively address the relevance of employing closed genomes in such comparative analyses. Indeed, they explicitly indicate that even genomes of low quality were not excluded from the study. Although I do not ask to remove these genomes in the analysis, appropriate consideration should be given to the possibility that the observed patterns may, to some extent, be influenced by genome fragmentation.

"In this analysis, it was possible to identify the existence of at least five orthoclusters (1, 2, 3, 4, and 12) that characterize one cluster (highlighted in green in Fig. 1)" - Genes within orthocluster 4, which delineates the *Bataviae*, *Shermani*, and *Tarassovi* serogroups, exhibit non-uniform distribution within these serogroups. Consequently, these genes cannot be utilized as discriminatory markers. Again, that could be an effect of genome fragmentation.

Clusters in Figures 1 and 2 are different. They should be harmonized concerning the representation of serogroups. For example:

Fig 1, cluster 1: Ballum, Javanica, Manhao, Ranarum, Sarmin, Mendeng.

Fig 2, cluster 1: Ballum, Javanica, Manhao, Ranarum, Canicola.

Fig 1, cluster 3: Pomona, Australis, Autumnalis, Cynopteri, Panama, Grippytyphosa, Djasiman.

Fig 2, cluster 3: Pomona, Australis, Autumnalis, Cynopteri, Panama, Grippytyphosa, Canicola.

Cluster 6, hereafter referred to as Class V, rfb synteny is shown for five of the eight serogroups depicted in Figure 1.

Figure 2 - There are bars indicating the syntenic blocks in samples from different clusters. It would be beneficial to specify that the coloring of these blocks is unrelated to the colors in the legend, as this could potentially lead to confusion. Perhaps consider providing a reference to explain the coloring. For example, are the syntenic blocks functionally related, as could be the case for the rfbCDBA cluster?

It is unclear whether the rfb synteny was conducted only with closed genomes and their selection. Additional clarification in the methodology section would be beneficial.

Figure 3 requires clarification. I presume that the green dashed line rectangle highlights proteins that differ in *L. icterohaemorrhagiae* compared to other classes and even species within the same class. However, there is no reference to the coloring (blue, red, white, pink). A legend is needed.

Twenty-six representative samples were chosen for the functional annotation of proteins in the rfb. What were the criteria for their selection as representatives? Are these criteria indicated in the supplementary information?

Figure 4D - I presume that the error bars take into account the number of representative strains from each class, the sum of which constitutes the 26 samples chosen for this analysis. While it is possible to count in Figure 4A how many samples were selected per group, it would be helpful to indicate that in the legend.

Finally, in the discussion it is mentioned: "Since we sought to emphasize the pathogenic group, the present study did not analyze in further detail the genetic composition of the rfb locus of class V". However, this class is included in all the analyses conducted. The authors should consider removing this class from the analyses if they intend to expand its examination in another publication, or, alternatively, they could provide a more general title, e.g.: Genetic structure and diversity of the rfb locus of species of genus *Leptospira*.

Reviewer #2 (Comments to the Authors (Required)):

The authors provide a timely, overarching analysis of the rfb locus of *Leptospirae*. This genus has been challenging microbiologists because of the extreme diversity of genotypes, phenotypes, host range, clinical outcomes etc. The rfb locus has been of particular interest because it pathogenetically and taxonomically critical proteins which are involved in biosynthesis of leptospiral LPS. In spite of the ever growing number of published leptospiral genomes, an unifying analysis of the rfb locus has been lacking. The authors suggest that the rfb locus defines 5 classes which do not strictly correspond to taxonomy according to species but, rather, shares serological properties. While this is not entirely unexpected, it does add to our understanding of relationships among *Leptospirae* in that it supports the notion (no by genomic data) that the cross-species and cross-strain serological nature of *Leptospiral* relationships is ultimately determined by genomic plasticity. I agree with the authors that the identification of syntenic blocks will enable identifying an isolate's serogroup by molecular means and lead to the development of better diagnostics / surveillance tools.

The paper is well written and the resolution and organization of the figures is good. My only concern with this study is that the data are not up to date. For instance, the studies by Ramli S et al. *Pathogens* 2022 and Senavirathna et al *Infect Genet Evol* 2023 include complete genomes and detailed analyses of the rfb locus which by and large support the findings presented herein. I recommend that the authors update their analyses using the genomes published in these two studies (as well as any other recent ones).

'Referee Cross-Comments: I agree with the other 2 reviewers' comments. In fact, reviewer 3 also suggests to include additional genomes (from Asian isolates). In fact, the Ramli and Senavirathna studies both feature Asian isolates.

Reviewer #3 (Comments to the Authors (Required)):

This article is discussing the genetic structure and diversity of the rfb locus in pathogenic species of the genus *Leptospira*. It's commendable that the authors have conducted a comprehensive analysis across different species. I would like to include isolates from the Asian region, as regional variations can provide additional insights into the genetic diversity of *Leptospira*.

have also noticed some spacing problems throughout the article

Abstract

Aim of the following sentence is not clear

Our findings can assist the development of new strategies based on molecular methods which can improve disease control.

Main antigen (not clear)

In the abstract the methods what they have used not mention clearly

In figure 4 A indicated not annotated. Is this refers to hypothetical protein

- Is the paper original? Yes
- Is it well written? Yes
- Are critical references given? Yes
- Is the length of the paper commensurate with the message? Yes
- Are all tables, figures, graphs, and photographs necessary? Yes
- If applicable, is "Material and Methods" section adequately written and referenced? Yes

Rebuttal Letter to manuscript #LSA-2023-02478-T

Dear editors and reviewers,

We greatly appreciate you for the time dealing with the manuscript and for the generous comments that allowed us to improve our work. We hope that all necessary corrections were made in the attached manuscript.

We would like to mention two main changes that were made to fit in with all the suggestions:

- 1) Inclusion of new *Leptospira* samples: we searched for new genomic data deposited in the NCBI database and included in our analysis. The search was carried out in December 2023;
- 2) The use of only pathogenic samples in the analyses: In this new analysis, we decided to focus only on samples grouped in P1 clade of *Leptospira*.

Considering these two points, the total number of samples used in this work was, coincidentally, 722 samples. As we are only dealing with pathogenic samples, we do not mention the class V (cluster 6) in the article as previously. In the next session of the document, we answer all the questions raised.

Best wishes,

Dr. Tetsu Sakamoto
Professor at Federal University of Rio Grande do Norte

On behalf of all authors.

Reviewer #1 (Comments to the Authors (Required)):

Genetic structure and diversity of the *rfb* locus of pathogenic species of genus *Leptospira*
By Ferreira et al.

The study conducted by Ferreira and colleagues aimed to comprehensively characterize the *rfb* locus of *Leptospira*, which encodes enzymes involved in O-antigen biosynthesis, from both a genetic and functional perspective. Utilizing a dataset comprising 722 genomes, the researchers performed a comparative analysis of gene composition within the *rfb* locus, resulting in the classification of samples into five distinct classes. Each class encompassed various serogroups sharing serological affinity.

The first part involved defining the distribution of orthologous genes from the *rfb* across the identified classes, revealing specific "orthogroups" that distinguished serogroups.

Subsequently, the researchers assessed synteny and gene composition within the *rfb* across the classes, proposing syntenic blocks as potential molecular markers. The study concluded with a functional characterization of proteins within superfamilies, providing insights into the representation of certain protein families across different classes. If the first part has already been published by others, the functional characterization is of particular interest, as it offers

valuable insights into specific modifications of LPS shared by groups of serogroups, distinguishing them from others.

Overall, the article is well-written and contributes significantly to the field, although certain aspects may benefit from additional explanation.

Comments:

- Statistics: One of the major comments is the lack of statistics to support the division in clusters or orthogroups.

Answer: We calculated the cluster stability score using the bootstrap approach for each cluster. This was performed using the function `clusterboot` implemented in the R library `fpc`. The procedures are detailed in the methodology section. The values of the support statistics are shown in Figure 1 and Figure 5.

- Methodology: a lot of details are missing see below.

- Recently, Chinchilla et al, PLOS NTD (10.1371/journal.pntd.0011733), proposed 4 clusters. Although they used less genomes how it compares to yours?

Answer: We became aware of the publication of Chinchilla et al. (2023) after submitting this paper. For this reason, our manuscript did not compare and discuss their results. By making this comparison, we did notice that there is a correspondence between the class I to IV proposed in this work and the 4 clusters proposed by Chinchilla et al. (2023). The only discordance between the works is the placement of the serogroup Pomona. In the Chinchilla et al. (2023), samples from serogroup Pomona are placed with samples of the class I, while in our work, they are placed in class II. We have added this fact to the Discussion section of the paper. (page 12-13)

- Page 4: "Recent studies... have identified 67 species so far and classified them into two major clades". There is no reference associated to this sentence, at present, there are 69 species, considering the inclusion of *L. sanjuanensis*, reported in Fernandes et al., 2022.

Answer: The references were incorporated and the number of species was updated in the manuscript (page 4).

In the initial part, 395 orthologous groups associated with the rfb were identified. Upon reviewing the methodology, the general approach employed to delineate these orthologous groups is described, utilizing as reference those genomes with an intact rfb locus, where *marR* was positioned at the 5' end and *DASS* at the 3' end. However, several details, such as the specific software employed and the associated parameters, are not explicitly disclosed and should be incorporated for comprehensive clarity.

Answer: We've added more details about this part of the methodology (page 16).

Figure 1 lacks clarity regarding the methodology employed for clustering orthologous genes. Although the text mentions the definition of a threshold, this is not included in the methodology section. Additional details would enhance the clarity of the analysis.

Answer: We have added more details about the procedures used to carry out the clustering analysis in the methodology (page 17-18).

In Figure 2, the orthoclusters are more evident, with distinct groups distinguished by coloration. Including a broad functional characterization, if feasible, for these orthoclusters might improve general comprehension.

Answer: Unfortunately, we were unable to find a feasible way of adding the functional information of the proteins in figure 2, but we have added some gene names at the top of the figure to serve as a reference.

"Most of the samples from the srg Icterohaemorrhagiae (n=117), srg Ballum (n=66), and srg Sejroe (n=46), which are the most frequent serogroups in the analysis, were grouped into clusters 1, 2, and 4, respectively" - Icterohaemorrhagiae is in cluster 2, Ballum in 1 and Sejroe in 4. The text should be modified.

Answer: The error has been corrected and we have revised the entire text to check that there is no inconsistency in the cluster names (page 6).

In Table 1, it is apparent that, despite most serogroups are predominantly clustering within a single group, certain genomes from specific serogroups exhibit clustering in more than one group. For example, Autumnalis: 2 genomes are in cluster 1 and, other 9 genomes in cluster 3. Is it attributed to genome fragmentation? The discussion surrounding genome fragmentation is entirely absent, and the researchers do not extensively address the relevance of employing closed genomes in such comparative analyses. Indeed, they explicitly indicate that even genomes of low quality were not excluded from the study. Although I do not ask to remove these genomes in the analysis, appropriate consideration should be given to the possibility that the observed patterns may, to some extent, be influenced by genome fragmentation.

Answer: We have analyzed in more detail those samples that do not follow the clustering pattern proposed in the paper. We wrote a paragraph in the discussion of our work on the possible reasons for this inconsistency in these few samples (page 13-14).

"In this analysis, it was possible to identify the existence of at least five orthoclusters (1, 2, 3, 4, and 12) that characterize one cluster (highlighted in green in Fig. 1)" -Genes within orthocluster 4, which delineates the Bataviae, Shermani, and Tarassovi serogroups, exhibit non-uniform distribution within these serogroups. Consequently, these genes cannot be utilized as discriminatory markers. Again, that could be an effect of genome fragmentation.

Answer: In fact, the orthocluster mentioned shows a non-uniform distribution among the samples of the Bataviae, Shermani, and Tarassovi serogroups. We rephrased this part in the manuscript and mentioned that the orthocluster has some orthogroups that are specific to this cluster, but not conserved in all samples of this cluster (page 7).

Clusters in Figures 1 and 2 are different. They should be harmonized concerning the representation of serogroups.

For example:

Fig 1, cluster 1: Ballum, Javanica, Manhao, Ranarum, Sarmin, Mendeng.

Fig 2, cluster 1: Ballum, Javanica, Manhao, Ranarum, Canicola.

Fig 1, cluster 3: Pomona, Australis, Autumnalis, Cynopteri, Panama, Grippotyphosa, Djasiman.

Fig 2, cluster 3: Pomona, Australis, Autumnalis, Cynopteri, Panama, Grippotyphosa, Canicola.

Cluster 6, hereafter referred to as Class V, rfb synteny is shown for five of the eight serogroups depicted in Figure 1.

Answer: Figures 1 and 2 have been altered in order to harmonize the serogroups represented.

Figure 2 - There are bars indicating the syntenic blocks in samples from different clusters. It would be beneficial to specify that the coloring of these blocks is unrelated to the colors in the legend, as this could potentially lead to confusion. Perhaps consider providing a reference to explain the coloring. For example, are the syntenic blocks functionally related, as could be the case for the *rfb*CDBA cluster?

Answer: We have rewritten part of the figure's legend, explicating that the colors of the bars do not correspond to the colors in the caption. We changed the color legend by adding arrows to avoid potential errors in the readers' interpretation of the figure.

It is unclear whether the *rfb* synteny was conducted only with closed genomes and their selection. Additional clarification in the methodology section would be beneficial.

Answer: The synteny analysis was carried out with the representative samples of each serogroup and only considering the *rfb* locus. We added more details to the methodology the procedures used to select the representative samples and to conduct the synteny analysis. (page 17-18)

Figure 3 requires clarification. I presume that the green dashed line rectangle highlights proteins that differ in *L. teterohaemorrhagiae* compared to other classes and even species within the same class. However, there is no reference to the coloring (blue, red, white, pink). A legend is needed.

Answer: We agree with the reviewer so we made some modifications in Figure 3 for better visualization and comprehension. We added on the top of the figure colored bars corresponding to the syntenic blocks highlighted in Figure 2. With this addition, we considered that the green dashed lines were not necessary, so we took off them. We also noted that the identity scale was missing, so we added it to the figure. We also added more information to the legend of the figure.

Twenty-six representative samples were chosen for the functional annotation of proteins in the *rfb*. What were the criteria for their selection as representatives? Are these criteria indicated in the supplementary information?

Answer: The representative samples from each serogroup chosen are the same as those shown in figure 2. We tried to select samples that had the best assembly status and that had the *rfb* locus intact. Unfortunately, not all serogroups had samples with an intact *rfb* locus (Sarmin and Djasiman). For these serogroups, we selected those samples that had the locus in the smallest number of fragments. Some details of the criteria used for their selection have been added to the text (page 17).

Figure 4D - I presume that the error bars take into account the number of representative strains from each class, the sum of which constitutes the 26 samples chosen for this analysis. While it is possible to count in Figure 4A how many samples were selected per group, it would be helpful to indicate that in the legend.

Answer: We added the number of samples used to make the error bars in each class in the legend.

Finally, in the discussion it is mentioned: "Since we sought to emphasize the pathogenic group, the present study did not analyze in further detail the genetic composition of the *rfb* locus of class V". However, this class is included in all the analyses conducted. The authors

should consider removing this class from the analyses if they intend to expand its examination in another publication, or, alternatively, they could provide a more general title, e.g.: Genetic structure and diversity of the rfb locus of species of genus *Leptospira*.

Answer: we agreed with the reviewer's suggestion and reworked the analysis considering only the pathogenic samples (P1 clade) of *Leptospira*.

Reviewer #2 (Comments to the Authors (Required)):

The authors provide a timely, overarching analysis of the rfb locus of Leptospirae. This genus has been challenging microbiologists because of the extreme diversity of genotypes, phenotypes, host range, clinical outcomes etc. The rfb locus has been of particular interest because it pathogenetically and taxonomically critical proteins which are involved in biosynthesis of leptospiral LPS. In spite of the ever growing number of published leptospiral genomes, an unifying analysis of the rfb locus has been lacking. The authors suggest that the rfb locus defines 5 classes which do not strictly correspond to taxonomy according to species but, rather, shares serological properties. While this is not entirely unexpected, it does add to our understanding of relationships among Leptospirae in that it supports the notion (no by genomic data) that the cross-species and cross-strain serological nature of Leptospiral relationships is ultimately determined by genomic plasticity. I agree with the authors that the identification of syntenic blocks will enable identifying an isolate's serogroup by molecular means and lead to the development of better diagnostics / surveillance tools. The paper is well written and the resolution and organization of the figures is good.

My only concern with this study is that the data are not up to date. For instance, the studies by Ramli S et al. Pathogens 2022 and Senavirathna et al Infect Genet Evol 2023 include complete genomes and detailed analyses of the rfb locus which by and large support the findings presented herein. I recommend that the authors update their analyses using the genomes published in these two studies (as well as any other recent ones).

'Referee Cross-Comments: I agree with the other 2 reviewers' comments. in fact, reviewer 3 also suggests to include additional genomes (from Asian isolates). In fact, the Ramli and Senavirathna studies both feature Asian isolates.

Answer: We agree with the reviewers and have added the recent *Leptospira* genomic data found in the database, including the genomes generated by the papers cited by the reviewer. The results obtained by adding these genomes did not alter the main conclusions of the paper.

Reviewer #3 (Comments to the Authors (Required)):

This article is discussing the genetic structure and diversity of the rfb locus in pathogenic species of the genus *Leptospira*. It's commendable that the authors have conducted a comprehensive analysis across different species. I would like to include isolates from the Asian region, as regional variations can provide additional insights into the genetic diversity of *Leptospira*.

Answer: We included more samples and applied the analyses carried out in this study. The results obtained by adding these genomes did not alter the main conclusions of the paper.

have also noticed some spacing problems throughout the article

Answer: We proofread the entire text to check and correct the problem noted by the reviewer.

Abstract

Aim of the following sentence is not clear

Our findings can assist the development of new strategies based on molecular methods which can improve disease control.

Main antigen (not clear)

In the abstract the methods what they have used not mention clearly.

Answer: We have rewritten the mentioned part of the abstract to clarify these points. We had to shorten even more the abstract to fit the journal requirement (175 words), so, unfortunately, we could not add more details about the methods in the abstract.

In figure 4 A indicated not annotated. Is this refers to hypothetical protein

Answer: The mention of unannotated proteins in figure 4A refers to proteins that have not been assigned by any domain in the Superfamily database. We modified the legend of the figure to clarify this. Instead of "annotated" and "not annotated", we wrote "with SCOP" and "without SCOP".

- Is the paper original? Yes
- Is it well written? Yes
- Are critical references given? Yes
- Is the length of the paper commensurate with the message? Yes
- Are all tables, figures, graphs, and photographs necessary? Yes
- If applicable, is "Material and Methods" section adequately written and referenced? Yes

March 7, 2024

RE: Life Science Alliance Manuscript #LSA-2023-02478-TR

Prof. Tetsu Sakamoto
Federal University of Rio Grande do Norte
Instituto Metr pole Digital
Campus Universit rio Central da UFRN - Av. Cap. Mor Gouveia, S/N - Lagoa Nova
Natal, Rio Grande do Norte 59078-900
Brazil

Dear Dr. Sakamoto,

Thank you for submitting your revised manuscript entitled "Genetic structure and diversity of rfb locus of pathogenic species of genus *Leptospira*". We would be happy to publish your paper in Life Science Alliance pending final revisions necessary to meet our formatting guidelines.

- please be sure that the authorship listing and order is correct
- please add the Twitter handle of your host institute/organization as well as your own or/and one of the authors in our system

A. FINAL FILES:

B. MANUSCRIPT ORGANIZATION AND FORMATTING:

****It is Life Science Alliance policy that if requested, original data images must be made available to the editors. Failure to provide**

original images upon request will result in unavoidable delays in publication. Please ensure that you have access to all original data images prior to final submission.**

The license to publish form must be signed before your manuscript can be sent to production. A link to the electronic license to publish form will be available to the corresponding author only. Please take a moment to check your funder requirements.

Sincerely,

Reviewer #3 (Comments to the Authors (Required)):

I have gone through the rebuttal submitted by the authors. I think they have addressed almost all of my comments. Therefore now this manuscript is ready for publication.

March 8, 2024

RE: Life Science Alliance Manuscript #LSA-2023-02478-TRR

Prof. Tetsu Sakamoto
Federal University of Rio Grande do Norte
Instituto Metr pole Digital
Campus Universit rio Central da UFRN - Av. Cap. Mor Gouveia, S/N - Lagoa Nova
Natal, Rio Grande do Norte 59078-900
Brazil

Dear Dr. Sakamoto,

Thank you for submitting your Resource entitled "Genetic structure and diversity of rfb locus of pathogenic species of genus *Leptospira*". It is a pleasure to let you know that your manuscript is now accepted for publication in Life Science Alliance. Congratulations on this interesting work.

DISTRIBUTION OF MATERIALS:

Again, congratulations on a very nice paper. I hope you found the review process to be constructive and are pleased with how the manuscript was handled editorially. We look forward to future exciting submissions from your lab.

Sincerely,
